# Edge-Based Self-Triggered Tracking Control for Stochastic Nonlinear Multiagent Systems

1st Wenzhe Wang

*College of Control Science and Engineering*

*Bohai University*

Jinzhou, China

2022008010@qymail.bhu.edu.cn

*Abstract*—This paper investigates the distributed tracking control problem for the stochastic multiagent systems with the asynchronous edge-based transmission mechanism. Different from the traditional time-based and event-based transmission mechanisms, the proposed self-triggered transmission mechanism can achieve the sampling task according to the current local information. In order to save the communication resources as economically as possible, the self-triggered transmission mechanism is designed as the asynchronous mode. The proposed self-triggered transmission mechanism can relieve the communication burden while reducing the complexity of the scheme. Finally, simulation verification is presented to demonstrate the effectiveness of the proposed control strategy.

*Index Terms*—Data transmission mechanism, stochastic multiagent systems, tracking control.

## I. INTRODUCTION

Stochastic multiagent systems (MASs) have been taken into consideration in many fields [1]. There are plenty of meaningful results to investigate cooperative control methods for stochastic MASs. For example, the leader-following control method of stochastic MASs in random vibration environment was studied in [2]. Ma *et al.* [3] investigated the consensus problem for a class of stochastic MASs. The distributed tracking control for the stochastic nonlinear MASs was considered both in [4] and [5]. Above mentioned works assume that the communication channels are ideal and each agent can receive the accurate information. This hypothesis is hardly achieved in real communication networks due to the limitation of transmission bandwidth.

Various event-triggered (ET)strategies have been proposed by an increasing number of researchers attracted by the wide applications of event-based control and also its aforementioned challenges. Until now, many works have focused on event-triggered control for various topics, including leaderless consensus, coordinated tracking, output feedback consensus, robustness consensus in the presence of external disturbances, etc. The ET mechanisms are often used to save communication resources [6]. As is known to all, the systems with ET mechanism must continuously check whether the trigger condition is satisfied. To address this problem, the self-triggered mechanism, one of ET mechanisms, has been considered in [7–9]. In particular, the neural tracking control was investigated in [10] based on the self-triggered mechanism. Considering that the transmission bandwidth is always limited, it brings the severe challenge to achieve the tracking task. Furthermore, reducing the complexity of the scheme is also the tough problem.

Taking inspiration from the above, this paper designs the adaptive tracking control strategy with the asynchronous edge-based transmission mechanism. The main contributions are listed below:

1) Different from the traditional event-based transmission mechanisms which have to set the trigger conditions, the proposed self-triggered transmission mechanism can achieve the sampling task according to the current local information.

2) In order to save the communication resources as economically as possible, the self-triggered transmission mechanism is designed as the asynchronous mode. Each communication channel is set with independent transmission mechanism to make the transmission be more flexible.

## II. PRELIMINARIES

### A. Problem Formulation

The exchanged information between agents is represented by a directed graph. In this paper, it has one leader labeled **0** and a set of $N$ followers which denoted 1 to $N$. Let $\mathcal{B} = \text{diag}\{b_1, ..., b_\mathcal{K}\} \in R^{\mathcal{K} \times \mathcal{K}}$ be the adjacency matrix of leader, and $b_i$ is the weight of the edge from $\mathcal{Y}_0$ to $\mathcal{Y}_i$.

The dynamic of the $i$th $(i = 1, \ldots, N)$ agent is defined in the strict-feedback form

$$\begin{cases} dx_{i,p} = (x_{i,p+1} + f_{i,p}(\bar{x}_{i,p}))dt + q_{i,p}(\bar{x}_{i,p})dw \\ dx_{i,n} = (u_i + f_{i,n}(\bar{x}_{i,n}))dt + q_{i,n}(\bar{x}_{i,n})dw \\ y_i = x_{i,1} \end{cases} \quad (1)$$

where $\bar{x}_{i,p} = [x_{i,1}, ..., x_{i,p}]$ represents the state vector with $p = 1, 2, ..., n-1$. $u_i$ is the control input signal, and the output signal of the $i$th agent is represented as $y_i$. The output of leader is denoted as $y_0$. $f_{i,p}(\bar{x}_{i,p})$, $f_{i,n}(\bar{x}_{i,n})$, $q_{i,p}(\bar{x}_{i,p})$ and $q_{i,n}(\bar{x}_{i,n})$ are unknown smooth nonlinear functions. $w$ denotes an $r$-dimension standard Brownian motion defined on the complete probability space $(\Xi, F, \{F_t\}_{t \geq 0}, P)$. $\Xi$ is a sample space. $F$ is a $\sigma$-field. $\{F_t\}_{t \geq 0}$ is a filtration, and $P$ is a probability measure.

## B. Asynchronous Edge-based Self-triggered Transmission Mechanism

Considering that determine when to transmit information from the leader to followers or between neighboring followers, an asynchronous edge-based self-triggered transmission mechanism is proposed in this paper. The mechanism consists of two kinds of transmission situations, designed correspondingly for the communication edges from the leader to informed followers, and the communication edges between neighboring followers.

In order to achieve the asynchronous edge-based self-triggered transmission mechanism, the intermediary control signals will be designed by utilizing the local information. Firstly, define the following local error variables as

$$\delta_0^i(t) = b_i(y_i - \breve{y}_0^i)$$
$$\delta_j^i(t) = a_{i,j}(y_i - \breve{y}_j^i) \tag{2}$$

where $\breve{y}_0^i$ and $\breve{y}_j^i$ are the signals on the edges $(0, i)$ and $(j, i)$, respectively. Then, two intermediary control signals are designed as follows:

$$\psi_0^i(t) = h_0^i(\delta_0^i)$$
$$\psi_j^i(t) = g_j^i(\delta_j^i) \tag{3}$$

where $h_0^i(.)$ and $g_j^i(.)$ are the designed continuous functions which are positively related to the variables $\delta^i$. Define the edge-based sampling errors as $\tau_r^i = y_r - \breve{y}_r^i$. Consider the asynchronous edge-based self-triggered transmission mechanism below

$$\breve{y}_r^i(t) = y_r(t_{r,k}^i), \forall t \in [t_{r,k}^i, t_{r,k+1}^i) \tag{4}$$

$$t_{r,k+1}^i = t_{r,k}^i + \frac{\eta_r |\psi_r^i| + m_r}{\max\{\hbar_r^i(t), |\zeta_r^i(t)|\}} \tag{5}$$

where $r = 0, 1, ..., i-1, i+1, ..., N$ and $t_{r,k}^i, t_{r,k+1}^i \in N^+$. Define $0 < \eta_r < 1$ and $0 < m_r < 1$. $y_r(t)$ is one of the output signals, and $\breve{y}_r^i(t)$ is the transmitted signal on the corresponding edge $(r, i)$. $\hbar_r^i(t)$ and $\zeta_r^i(t)$ are the continuous functions, which are designed below

$$\zeta_r^i(t) = \frac{y_r(t_{r,k}^i) - \breve{y}_r^i(t)}{t_{r,k} - t}$$
$$\hbar_r^i(t) = v_r e^{\frac{\delta_r^i}{2\iota_r}} \tag{6}$$

where $v_r$ and $\iota_r$ are positive constants. $\zeta_r^i(t)$ is designed to describe the change degree of intermediary interval. $\hbar_r^i(t)$ is designed to effectively adjust the trigger interval because it has the positive correlation with the synchronization errors.

When the trigger procedure begins and condition (5) is satisfied, the transmitted signal $\breve{y}_r^i(t)$ will be assigned as the output signal $y_r(t_{r,k}^i)$. Then, the value will remain constant over the period of time $t \in [t_{i,k}, t_{i,k+1})$. Moreover, the next trigger time $t_{i,k+1}$ will be got.

*Remark 1*: The Zeno behaviors are discussed below. Considering the equation (5), the next trigger instant $t_{r,k+1}^i$ is directly obtained based on the current trigger instant $t_{r,k}^i$ with the

calculation. It is obvious that the trigger intervals are not equal to zero at any time. Hence, the Zeno behaviors are avoided.

Combined with the tracking task in this paper, the tracking errors is denoted by the following equation

$$s_{i,1} = \sum_{j=1}^{N} a_{i,j}(y_i - y_j^i) + b_i(y_i - y_0^i)$$
$$= \sum_{j=1}^{N} a_{i,j}(y_i - y_j) + b_i(y_i - y_0) - (\sum_{j=1}^{N} a_{i,j} + b_i)\tau_r^i \tag{7}$$

## III. CONTROLLER DESIGN

First, define the unknown constant $\xi_i$ as

$$\xi_i = \max\left\{||\varrho_{i,m}||^2\right\}, \quad m = 0, 1, ..., n \tag{8}$$

The estimation of $\xi_i$ is defined as $\hat{\xi}_i$, and there exists the estimation error $\tilde{\xi}_i$ such that $\tilde{\xi}_i = \xi_i - \hat{\xi}_i$.

Define the following transformation

$$s_{i,k} = x_{i,k} - \alpha_{i,k-1} \tag{9}$$

where $\alpha_{i,k-1}$ is the virtual control signal.

Based on the (7), (8) and (9), the asynchronous edge-based self-triggered control protocol is designed as follows:

$$\alpha_{i,1} = \frac{1}{b_i + d_i}\left[-c_{i,1}s_{i,1} - \sum_{j=1}^{N} a_{i,j}x_{j,2} + b_i\dot{y}_0 \right.$$
$$\left. - \frac{\xi_{i,1}}{2v_{i,1}^2}s_{i,1}^3\phi_{i,1}^T\phi_{i,1}\right] - \frac{3}{4}s_{i,1}$$

$$\alpha_{i,k} = -c_{i,k}s_{i,k} - \frac{9}{4}s_{i,k} - \frac{1}{4}\varpi_{i,k}s_{i,k} + H_{i,k20} \tag{10}$$
$$- \frac{\hat{\xi}_{i,k}}{2v_{i,k}^2}s_{i,k}^3\phi_{i,k}^T\phi_{i,k}$$

$$u_i = -c_{i,n}s_{i,n} - \frac{\hat{\xi}_{i,n}}{2v_{i,n}^2}s_{i,n}^3\phi_{i,n}^T\phi_{i,n} + H_{i,n20}$$

where $v_{i,.}$ $c_{i,.}$ and $\aleph_{i,.}$ are positive constants. $H_{i,.20}$ are the output signals of the second-order sliding mode integral filter, and $\varpi_{i,k}$ are defined as

$$\varpi_{i,k} = \begin{cases} b_i + d_i, & k = 2 \\ 1, & otherwise \end{cases}$$

Then, the corresponding adaptive laws are selected as

$$\dot{\hat{\xi}}_{i,1} = \frac{1}{2v_{i,1}^2}s_{i,1}^6 - \sigma_{i,1}\hat{\xi}_{i,1}$$

$$\dot{\hat{\xi}}_{i,k} = \frac{1}{2v_{i,k}^2}s_{i,k}^6 - \sigma_{i,k}\hat{\xi}_{i,k} \tag{11}$$

$$\dot{\hat{\xi}}_{i,n} = \frac{1}{2v_{i,n}^2}s_{i,n}^6 - \sigma_{i,n}\hat{\xi}_{i,n}$$

*Theorem 1*: Considering the stochastic MASs (1) under the asynchronous edge-based self-triggered transmission mechanism (4) and (5), all the signals, especially tracking errors (7),

are bounded in probability under the action of Assumptions 1 and 2, input control signals (10), and adaptive laws (11).

*Proof 1:* Based on the backstepping framework in stochastic MASs, consider the following transformation

$$s_{i,1} = \sum_{j=1}^{N} a_{i,j}(y_i - \breve{\gamma}_j^i) + b_i(y_i - \breve{y}_0^i) \tag{12}$$

$$s_{i,k} = x_{i,k} - \alpha_{i,k-1} \tag{13}$$

where $\alpha_{i,k-1}$ is the virtual control signal. the process of controller design is displayed as follows:

**Step 1:** From (8), one has

$$ds_{i,1} = [(b_i + d_i)(s_{i,2} + \alpha_{i,1} + f_{i,1}(x_{i,1})) - \sum_{j=1}^{N} a_{i,j}(x_{j,2}$$
$$+ f_{j,1}(x_{j,1}))) - b_i \ddot{y}_0]dt + [(b_i + d_i)q_{i,1}(x_{i,1})$$
$$- \sum_{j=1}^{N} a_{i,j}q_{j,1}(x_{j,1})]dw \tag{14}$$

Select the Lyapunov function as

$$V_{i,1} = \frac{1}{4}s_{i,1}^4 + \frac{1}{2}\widetilde{\xi}_{i,1}^2 \tag{15}$$

Based on (10) and (11), one has

$$LV_{i,1} \le -c_{i,1}s_{i,1}^4 - \frac{\sigma_{i,1}}{2}\widetilde{\xi}_{i,1}^2 + \frac{1}{4}(b_i + d_i)s_{i,2}^4 + \Delta_{i,1} \tag{16}$$

where $\Delta_{i,1} = \frac{1}{4}\varepsilon_{i,1}^4 + \frac{\sigma_{i,1}}{2}\xi_{i,1}^2$.

**Step $k$ ($2 \le k \le n-1$):** From (15), one has

$$ds_{i,k} = (x_{i,k+1} + f_{i,k} - L\alpha_{i,k-1})dt$$
$$+ (q_{i,k} - \sum_{j=1}^{k-1} \frac{\partial \alpha_{i,k-1}}{\partial x_{i,j}} q_{i,j})dw \tag{17}$$

Choose the Lyapunov function as

$$V_{i,k} = V_{i,k-1} + \frac{1}{4}s_{i,k}^4 + \frac{1}{2}\widetilde{\xi}_{i,k}^2 \tag{18}$$

Based on (10) and (11), one has

$$LV_{i,k} \le - \sum_{m=1}^{k}(c_{i,m}s_{i,m}^4 + \frac{\sigma_{i,m}}{2}\widetilde{\xi}_{i,m}^2) + \frac{1}{4}s_{i,k+1}^4 + \Delta_{i,k}$$
$$- \frac{\iota_{i,1}}{2}\widetilde{h}_{i,1}^2 - \frac{\jmath_{i,1}}{2}\widetilde{\gamma}_{i,1}^2 \tag{19}$$

where $\Delta_{i,k} = \Delta_{i,k-1} + \frac{1}{4}\varepsilon_{i,k}^4 + \frac{\sigma_{i,k}}{2}\xi_{i,k}^2 + \frac{1}{4}H_{i,km}^4 + \frac{3}{4}z_k^2$.

**Step $n$:** Then, one has

$$ds_{i,n} = dx_{i,n} - d\alpha_{i,n-1}$$
$$= (u_i + f_{i,n} - L\alpha_{i,n-1})dt - (\sum_{m=1}^{n-1} \frac{\partial \alpha_{i,n-1}}{\partial \hat{x}_{i,m}} q_{i,m})dw \tag{20}$$

Select the Lyapunov function as

$$V_{i,n} = \frac{1}{4}s_{i,n}^4 + \frac{1}{2}\widetilde{\xi}_{i,n}^2 + V_{i,n-1} \tag{21}$$

Based on (10)and (11), one has

$$LV_{i,n} \le - \sum_{m=1}^{n}(c_{i,m}s_{i,m}^4 + \frac{\sigma_{i,m}}{2}\widetilde{\xi}_{i,m}^2) + \frac{\sigma_{i,n}}{2}\xi_{i,n}^2$$
$$+ \frac{3}{4}\jmath_n^2 + \frac{1}{2}v_{i,n}^2 + \frac{1}{4}\delta_{i,n}^4 + \frac{1}{4}H_{i,nm}^2 + \Delta_{i,n-1} \tag{22}$$

In view of (22), one has

$$LV_{i,n} \le -\aleph V_{i,n} + \flat \tag{23}$$

where $\aleph = \min\{2c_{i,m}, \iota_{i,1}, \jmath_{i,1}, \sigma_{i,m}\}$ and $\flat = \Delta_{i,n_i-1} + \frac{\sigma_{i,n}}{2}\xi_{i,n}^2 + \frac{3}{4}\jmath_n^2 + \frac{1}{2}v_{i,n}^2 + \frac{1}{4}\delta_{i,n}^4 + \frac{1}{4}H_{i,nm}^2$. The following inequation can be got

$$\frac{dE(V(t))}{dt} \le -\aleph E(V(t)) + \flat, \quad t \ge 0 \tag{24}$$

Then, one obtains

$$0 \le E[V(t)] \le e^{-\aleph t}(V(0) - \frac{\flat}{\aleph}) + \frac{\flat}{\aleph} \tag{25}$$

All signals in MASs (1) are SGUUB in probability.

## IV. SIMULATION VERIFICATION

In this section, a numerical example is utilized to verify the effectiveness of control scheme. Consider a class of stochastic MASs with four followers and one leader. The adjacency matrix $\mathcal{A}$ is written as

$$\mathcal{A} = \begin{bmatrix} 0 & 0 & 0 & 1 \\ 1 & 0 & 0 & 0 \\ 0 & 1 & 0 & 0 \\ 0 & 1 & 1 & 0 \end{bmatrix} \tag{26}$$

The dynamic of the $i$th ($i = 1, 2, 3, 4$) agent is defined in the strict-feedback form

$$\begin{cases} dx_{i,1} = (x_{i,2} + \sin(x_{i,1}))dt + \cos(x_{i,1})dw \\ dx_{i,2} = (u_i + \sin(x_{i,2}))dt + \cos(x_{i,1})dw \\ y_i = x_{i,1} \end{cases} \tag{27}$$

Then, the dynamic of the leader is modeled as $y_0 = \sin(t)$.

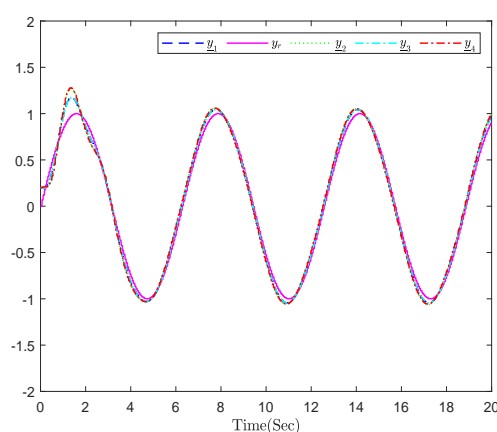

Fig. 1. Tracking performance.

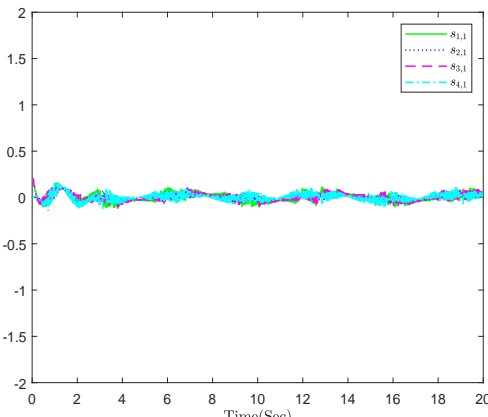

Fig. 2. Tracking errors.

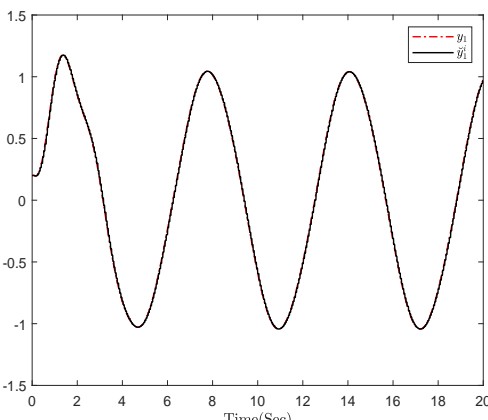

Fig. 3. Trigger performance.

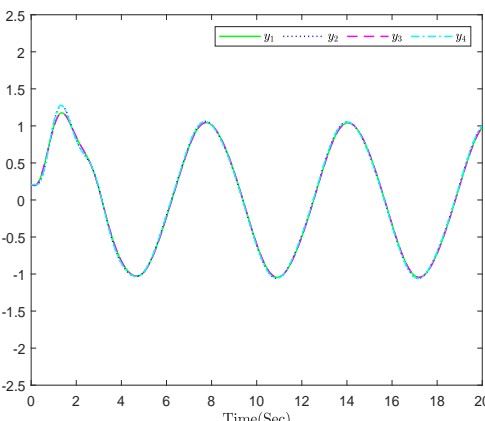

Fig. 4. State evolution.

Fig. 1 illustrates the tracking performance of proposed control method. The tracking errors are illustrated in Fig. 2. The trigger performance is shown in Fig. 3. In addition, the original state evolution is shown Fig. 4 to verify the control performance.

## V. CONCLUSION

In this paper, the asynchronous edge-based self-triggered transmission mechanism has been investigated. The burden of the transmission has been reduced. Eventually, simulation verification have been provided to testify the effectiveness of the proposed control strategy.

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
