# OpenReview forum: "Edge-Based Self-Triggered Tracking Control for Stochastic Nonlinear Multiagent Systems"
_IEEE.org/ICIST/2024/Conference — IEEE ICIST 2024 Conference Submission_

### Official Review · Reviewer_ejnB · 2024-08-20
**Edge-Based Self-Triggered Tracking Control for Stochastic Nonlinear Multiagent Systems**

**Rating:** 2
**Confidence:** 5

**Review:**

Thank you for submitting your manuscript titled "Edge-Based Self-Triggered Tracking Control for Stochastic Nonlinear Multiagent Systems" to IEEE ICIST. After careful consideration and review, I regret to inform you that I must recommend the rejection of your paper. Below are the key reasons for this decision:
1.The paper does not sufficiently demonstrate the novelty of the proposed method or its contribution to the field. Many aspects of the proposed approach appear to be similar to existing methods and lack a clear advancement in the state of the art.

---

### Official Review · Reviewer_thva · 2024-08-22
**Edge-Based Self-Triggered Tracking Control for Stochastic Nonlinear Multiagent Systems**

**Rating:** 5
**Confidence:** 5

**Review:**

The article addresses the distributed tracking control problem for stochastic nonlinear multi-agent systems (MASs) using an edge-based self-triggered transmission mechanism. Unlike traditional time-based or event-based transmission methods, the proposed self-triggered mechanism allows sampling based on current local information, thereby reducing communication resource usage. The mechanism operates asynchronously, meaning each communication channel operates independently, enhancing flexibility and reducing communication overhead. The effectiveness of the proposed control strategy is validated through simulations, demonstrating its potential to improve the performance of stochastic MASs in various applications. On the whole, there are still some problems that need to be corrected.

Comments:

1.The article introduces an innovative self-triggered transmission mechanism, but the explanation of how it distinctly differs from existing methods could be expanded for better clarity.

2.The description of the proposed method is detailed, but it could be enhanced by including a flowchart or a pseudocode example to help readers better understand.

3.The picture in the simulation experiment is not high in definition, and the relevant explanation is not enough.

4.The research content is well constructed. But there are grammatical errors and incorrect English writing in the current manuscript. Please make the necessary corrections.

---

### Official Review · Reviewer_GNs4 · 2024-08-22
**Edge-Based Self-Triggered Tracking Control for Stochastic Nonlinear Multiagent Systems**

**Rating:** 6
**Confidence:** 5

**Review:**

This paper investigates the distributed tracking control problem for the stochastic multiagent systems with the asynchronous edge-based transmission mechanism. This work is well organized. Below are some comments.
(1)	The contributions should be illustrated in a clearer manner. For example, what is the main improvement of the paper compared to the existing results. The authors should explain the unique contributions of this paper.
(2)	I think that adding a block diagram/scheme will help to understand how your method works.
(3)	It is better to give a guideline of selection for all control parameters.
(4)	The simulation results should be explained more carefully.
(5)	The paper is well presented, spelled correctly. I recommend authors to carefully read the entire paper to find possible misspellings.

---

### Decision · Program_Chairs · 2024-09-08

Reject